# Predicted IL-18/IL-18R Binding Improvement Through Protein Interface Modification with Computer-Aided Design

**DOI:** 10.3390/biom15101360

**Published:** 2025-09-25

**Authors:** Napat Prompat, Chariya Peeyatu, Jirakrit Saetang, Niran Roongsawang, Surasak Sangkhathat, Varomyalin Tipmanee

**Affiliations:** 1Department of Biomedical Sciences and Biomedical Engineering, Faculty of Medicine, Prince of Songkla University, Songkhla 90110, Thailand; napat.pr@psu.ac.th (N.P.); chariya.pyt@gmail.com (C.P.); 2Faculty of Medical Technology, Prince of Songkla University, Songkhla 90110, Thailand; 3International Center of Excellence in Seafood Science and Innovation, Faculty of Agro-Industry, Prince of Songkla University, Hat Yai, Songkhla 90110, Thailand; jirakrit.s@psu.ac.th; 4Microbial Cell Factory Research Team, National Center for Genetic Engineering and Biotechnology, National Science and Technology Development Agency, Pathum Thani 12120, Thailand; niran.roo@biotec.or.th; 5Department of Surgery, Faculty of Medicine, Siriraj Hospital, Mahidol University, Bangkok 10700, Thailand; s.sangkhathat@gmail.com

**Keywords:** cytokine-mediated immunotherapy, interleukin-18, molecular dynamics simulation, structure-guided design

## Abstract

Cytokine-mediated immunotherapy has rapidly emerged as an effective alternative approach for cancer treatment by modulating the anti-tumor response. Interleukin-18 (IL-18) is considered as a promising cancer therapeutic agent due to the ability of cytokines to inhibit cancer by enhancing natural killer (NK) cell and cytotoxic T cell responses. Since the activity of IL-18 is required for the specific binding to IL-18 receptors, the modification of binding residue at the protein interface is an attractive strategy for IL-18 activity enhancement. The aim of this study was to design and predict mutations increasing the activity of IL-18 through computational structure-based energy calculation and molecular dynamic simulations. Four candidate mutations, E6M, E6M+N111S+R131G, E6M+K129M+R131G, and E6M+N111S+K129M+R131G, could affect/facilitate the receptor binding and stability compared to the wild-type via electrostatic interaction. MD simulations demonstrated that the predicted mutation on IL-18 had no influence on the overall conformation stability, but increased flexibility in the β8–β9 hairpin loop. Furthermore, the dynamic behavior suggested that these candidates could be an alternative for the improvement of IL-18 biological activity, though the full simulation of the IL-18 complex remains necessary. In summary, this study offered a computer-aided design strategy which was of beneficial use in the design and development of IL-18 to increase its cytokine potency and efficiency.

## 1. Introduction

Cancer is a major cause of global death [1]. The World Health Organization (WHO) has reported an increase of 18.1 million cases and an estimated death rate of 9.6 million [1]. An increase in cancer patients has driven the development of new and more effective treatments [2]. Immunotherapy, immune-based treatment that exploits a patient’s immune system to recognize and eliminate cancer, has emerged as a promising approach for cancer therapy and shown its great potential for treating human cancers due to selectivity and long-lasting effects [3]. The field of immunotherapy shows great progress toward a cancer treatment, such as immune checkpoint blockage agents, monoclonal antibodies, cancer vaccines, oncolytic viruses, adoptive cell therapy, and cytokine-mediated immunotherapy [4].

Interleukin-18 (IL-18), an 18 kDa immunostimulatory cytokine primarily secreted by macrophages, is encoded by the IL-18 gene on chromosome 11q3.1 [5]. This cytokine plays important roles in anti-tumor immune responses, including the activation and proliferation of T cells, the induction of IFN-γ production in natural killer (NK) cells, and the regulation of several cytokines in both innate and adaptive immunity against cancer cells, leading to the enhancement of the patient’s immune responses and tumor regression [6].

In addition, the binding modes of IL-18 and the IL-18 receptor (IL-18R) show the crucial roles of the biological function of IL-18. The crystal structure [7] revealed that binding sites of IL-18 are composed of two parts. The first part of the binding site, interacting specifically with IL-18Rα, includes site I and site II, and site III is important for IL-18Rβ receptor binding. Some amino acid residues, including Glu-6, Lys-53, and Ser-55, that are located on the binding interface between IL-18 and the specific receptor importantly serve as binding residues to the formation of a complex on the surface of the receptor [8].

The modification of amino acids on the binding sites of IL-18 through molecular cloning and mutagenesis techniques can enhance the biological activity regarding the induction of IFN-γ production in tumor immunotherapy [9]. Interestingly, the previous studies exhibited that the engineered IL-18 with E6A, E6K, and E6K+T63A mutations have shown ability to induce IFN-γ production in NK cell lines [9,10,11]. The results suggested that Glu-6 is a critical amino acid residue for IL-18 biological activity enhancement [9,12,13]. Besides this, surface cysteine modification can also enhance IL-18 function via reducing protein aggregation [14]. Although the activity of those IL-18 mutants was increased, they did not test any other possible amino acid residues, nor did they examine the effects of the mutations on the biological activity of IL-18; moreover, it is expensive and time-consuming to produce recombinant engineered proteins with in vitro studies.

Nowadays, computational design and in silico screening generate novel engineered proteins with reduced related costs and time spent on the workbench. Therefore, we apply comprehensive computational approaches using molecular dynamics (MD) simulation to examine the effects of Glu-6 mutation on the biological activity of IL-18, so that it will be provided with a basis and advantages as a structure-guided protein design for further exploitation of cytokine-mediated immunotherapy.

## 2. Materials and Methods

### 2.1. In Silico Mutagenesis and Energy Decomposition

An X-Ray structure of human IL-18 in complex with IL-18 receptor alpha, with a resolution of 3.10 Å (PDB entry: 3WO3) [7], was downloaded from Research Collaboratory for Structural Bioinformatics (RCSB) Protein Data Bank [15]. The 3WO3 was chosen in this study as it is the only experimental IL-18 structure bound to the IL-18 receptor. The solvent molecules in the co-crystallized protein structure were removed. Chain A and chain B were selected for further analysis.

The protein structure visualization and rendering were performed using Visual Molecular Dynamics (VMD) version 1.9.3 to determine the information about atom types, numbering of atoms, masses, charges, bonds, angles, and dihedrals of the complex structure [16]. Prior to the generation of the mutated structure, the 3WO3 structure must be subjected to an optimization procedure using the RepairPDB command in FoldX5 [17] to remove steric hindrances and unfavorable high-energy conformations.

To generate the mutant protein, the selected amino acid of the repaired structure was modified into the new residues using the BuildModel function. The numberOfRuns option was set to 5 for ensuring minimal energy conformation because some residues can have different rotameric conformations. All other parameters were set to default, including pH (7), ionic strength (0.05 M), vdwDesign (2), and temperature (310 K). The contribution of individual residue to the binding energy of IL-18 towards its receptor was calculated using per-residue energy decomposition (ΔG_residue_).

The crystal structure of human IL-18 in complex with a receptor was decomposed by the Fast Fourier Transform (FFT)-based docking method implemented in the pyDockEneRes server [18]. The residue of interest of the IL-18 structure was mutated to all 19 possible amino acids using BuildModel module in the FoldX algorithm. To validate the computational predictions, the reported mutations with in vitro activities were also generated using the BuildModel function, as described above. The 17 known mutations, related function impairment or IL-18 activity improvement [8,9], were considered for method validation, namely, K4A, L5A, K8A, D17A, R13A, M33A, D35A, R58A, M60A, R104A, D132A, K79A, K84A, D98A, E6K, and E6K+T63A.

### 2.2. Analysis of Relative Binding Free Energy Change upon Mutation

AnalyzeComplex function of FoldX software was used to calculate the free energy change after IL-18 was mutated with respect to the IL-18 receptor (IL-18R). Three types of free energy changes (ΔΔG and ΔG_mutant_ − ΔG_wild-type_) were computed. The difference in binding free energy (ΔΔG_binding_) of the IL-18/IL-18R complex determined the change in interaction. The difference in folding free energy (ΔΔG_folding_) of IL-18 estimated the effect of mutations on the stability of the IL-18/IL-18R complex.

The difference in free energy folding of free IL-18 (ΔΔG_folding of free IL-18_) calculated the sole influence of mutations on the IL-18 stability. Correlations between the relative binding free energy changes upon mutation (ΔΔG) of the IL-18/IL-18R complex were evaluated using Pearson’s correlation and simple linear regression. A *p* value < 0.01 and 0.05 were considered statistically significant. All statistical analyses were performed using Libre Office Calc 5 in Ubuntu 16.04 LTS Linux systems.

### 2.3. Molecular Dynamics (MD) Simulation

MD simulations were performed to mimic an in vivo condition and investigated for conformational dynamic behaviors. All MD simulations were conducted using AMBER 16 software [19]. The crystal structure of human IL-18, PDB code 3WO3, and designed IL-18 mutants from 2.3 were chosen for the initial structures for preparing the MD simulation. The protonation state of the ionizable amino acids (Lys, Arg, His, Asp, and Glu) were calculated using PROPKA webtools [20] at pH 7.

All missing hydrogen atoms of the protein structure were added using the LEaP module in AMBER16 package. Each system was neutralized by the counter ions and solvated in a concentration of 0.15 mol dm-3 sodium chloride (NaCl) using a TIP3P [21] water box with an extended distance of 14 Å from the protein surface. Finally, the system contained approximately 11,400 water molecules and 30 NaCl pairs, excluding counter ions. The system was then saved into an AMBER parameter topology (.prmtop) file along with an AMBER7 restart file (.rst7) for energy minimization.

The energy minimization (imin = 1) was carried out by the steepest descent and conjugate gradient methods for 1000 and 1000 steps, respectively, to remove bad van der Waals contact, with a cutoff 16 Å (cut = 16). The energy minimization methods were implemented in the PMEMD.cuda module under a periodic boundary condition (ntb = 1). The final minimized coordinate was obtained in the form of an AMBER7 restart file (.rst7) to later continue the MD simulation.

The minimized protein structure was subjected to perform the MD simulation (imin = 0) in 2-nanosecond isothermal ensemble (NVT) simulation with a 1fs time step, using force constants of 200, 100, 50, 25, and 10 kcal/mol, respectively. The harmonic restrained potential was applied into the protein residue (Residue 1 to 156). Each force constant lasted for 400 picoseconds (ps). The temperature of 310 K (37 °C) was controlled with Langevin dynamics (ntt = 3) [20].

The nonbonded and electrostatic interactions were handled with a cutoff of 16 Å using Particle Mesh Ewald (PME) summation. The NVT ensemble was carried out with the PMEMD.cuda module under a periodic boundary condition (ntb = 1). The hydrogen-connected bond (H-X) was restrained using the SHAKE algorithm (ntc = 2) [22]. Then, the system was simulated under the periodic boundary condition (ntb = 1) with the isothermal–isobaric (NPT) ensemble of 310 K (temp0 = 310) and 1 atm (1.013 bar) (pres0 = 1.013) until reaching 150 nanoseconds (ns) (2500 snapshots). The temperature and pressure were regulated using a weak-coupling (Berendsen) algorithm (ntt = 1 and ntp = 1) [23].

The hydrogen-connected bond was restrained using the SHAKE algorithm (ntc = 2). The first 90 ns simulation (1500 snapshots) was omitted, and the MD trajectory was extracted from the last 60 ns (1000 snapshots) for a structural analysis using the cpptraj module in AMBER16. The structure visualization was performed using VMD software.

### 2.4. Trajectory Analysis

Trajectory analysis was performed to assess the structural and dynamic properties of IL-18 and its mutated variants during molecular dynamics (MD) simulations. Root mean square deviation (RMSD) was calculated to measure the average distance between backbone atoms of the protein structure from MD snapshots and the reference crystal structure, serving as an indicator of structural similarity and energy convergence (stability), with values typically acceptable below 3 Å [24].

The root mean square fluctuation (RMSF) was used to evaluate the flexibility of individual residues by measuring spatial deviations over the trajectory; the RMSF value, expressed in angstroms (Å), was calculated based on the fluctuation of atomic positions relative to their average [25,26].

Conformational differences between mutated and native IL-18 were further assessed by computing the distance pattern, defined as the average distance of each amino acid’s backbone geometric center (N, C, CA, and O atoms) to the origin point (0,0,0) across 1000 MD snapshots. To circumvent translation/rotation, the MD trajectories were aligned with the IL-18 crystal backbone structure (3WO3), allowing for differences in these distances to clearly reflect structural changes. Furthermore, the time-dependent RMSD of each amino residue relative to the IL-18 crystal backbone structure was plotted to show the conformation change in greater detail.

Additionally, hydrogen bonding analysis was carried out using the same set of 1000 trajectory snapshots, with hydrogen bonds identified based on a donor–acceptor distance of ≤3.5 Å and a donor–hydrogen–acceptor angle of ≥150°, in accordance with a previous study [27].

## 3. Results and Discussion

### 3.1. Per-Residue Decomposition Energy

The per-residue decomposition energy was introduced to investigate each amino acid contributing to the binding energy. A technique of per-residue binding free energy decomposition can identify the favorable or unfavorable residue on IL-18/IL-18R. The technique can not only easily facilitate the point-by-point amino acid alteration to predict the binding energy, but also quantify the preference of amino acids toward IL-18/IL1-18R interaction. The per-residue binding free energy relied on the Fast Fourier Transform (FFT)-based docking method.

The pyDockEneRes server was used to elucidate the important residues of IL-18 that contributed to the receptor binding. The results are shown in Appendix A. The residues with negative binding free energy contributions are identified as key (favorable) residues responsible for the IL-18/IL-18 receptor interactions at the interface. On the other hand, the residue with positive binding free energy values suggest unfavorable interaction residues on the binding interface of IL-18.

There were 11 important residues from binding site I and 7 residues from binding site II of IL-18 with binding free energies less than −1 kcal/mol. The 11 residues of binding site I comprise Asp17, Phe21, Glu31, Asp32, Thr34, Asp35, Asp37, Asp40, Asp132, Leu133, and Phe134, whereas the 7 residues of binding site II were Tyr1, Leu5, Lys8, Lys53, Ser55, Met60, and Gln103 (Appendix A).

Seven residues of site I, namely, Asp17, Phe21, Asp32, Asp37, Asp40, and Phe134, and one residue of site II, Met60, contribute −3.0 to −5.0 kcal/mol of free energy in the IL-18/IL-18 receptor complex. Some amino acid residues, such as Met33, Lys93, and Arg104, had values of binding free energy of approximately −1.0 kcal/mol. These residues contributed favorably to form a ligand–receptor complex by direct binding to the IL-18Rα. Similarly, in binding site III, the contributions from Lys79, Lys84, and Asp98 were −0.08, −0.21, and 0.32 kcal/mol, respectively, in IL18/IL18Rβ. In addition, analysis of the binding free energies based on the electrostatic contribution and the van der Waals energy revealed that the interaction between IL-18 and its receptor is driven by favorable electrostatic and van der Waals contributions that are compensated by unfavorable desolvation contributions (Appendix A).

The results reasonably agreed with the previous experimental mutagenesis studies [10,22] that the alanine mutation of these residues could abolish the receptor binding and leads to the decreased function of the IL-18. Thus, the biological activity of IL-18 required the specific binding to IL-18 receptors, supporting the importance of the residues with the favorable energetic contributions at interface binding, which play a critical role in IL-18/IL-18R interaction and biological function. Analysis of the per-residue free energy decomposition also revealed unfavorable interface residues.

The result exhibited that 11 residues at the binding interface of IL-18 were identified as unfavorable residues due to their positive contribution of binding free energies (Figure 1). The four unfavorable residues of binding site I include Asn41, Arg44, Lys129, and Arg 131, and the seven residues of site II consist of Glu6, Asp54, Gln56, Asp110, Asn111, Met113, and Asn155. Since the presence of energetically favorable residues is necessary for IL-18 binding, substituting unfavorable interfacial residues with favorable ones should improve the binding affinity and biological activity of IL-18.

### 3.2. Validation of the Computational Approach

To test the potential reliability of the computational method prior to using this approach to screen favorable amino acid substitutions for IL-18 activity enhancement, a total 17 reported mutants in human IL-18 from experiments [8,9,10] were chosen for a relative binding energy evaluation using the FoldX algorithm, namely, K4A, L5A, E6A, E6K, K8A, D17A, R13A, M33A, D35A, R58A, M60A, R104A, D132A, K79A, K84A, D98A, and E6A+T63A. These selected IL-18 mutants were in comparison with in vitro functional studies, as shown in Appendix A.

The activity of the wild-type was taken as standard (100%) for the calculation of relative activity. The relative activity of the mutant was determined by a ratio of a reported activity from mutant IL-18 to an activity from wild-type IL-18.$$\mathrm{Relative}\;\mathrm{activity}=\frac{In\;vitro\;activity\;of\;mutant\;(\%)}{In\;vitro\;activity\;of\;wild-type\;(\%)}$$

The relative free energy (ΔΔG) values obtained by FoldX were calculated based on the difference in the energy of the mutant (ΔG_mutant_, in kcal/mol) relative to the energy of the wild-type (ΔG_wild-type_, in kcal/mol) of IL-18.

ΔΔG = ΔG_mutant_ − ΔG_wild-type_

Three types of ΔΔG values were categorized into

(1)“ΔΔG_binding_”, which estimated the change in the binding affinity of the wild-type and the mutant variant toward its receptor;(2)“ΔΔG_folding of complex_”, which determined the impact of mutations on the stability of the IL-18/IL-18R complex;(3)“ΔΔG_folding of free IL-18_”, which investigated the effect of mutations on the stability of isolated IL-18.

ΔΔG < 0 (i.e., negative) indicated favorable stability and affinity, and promising mutations, whereas ΔΔG > 0 (i.e., positive) indicated unfavorable stability and affinity toward its receptor.

In comparison with experimental data, three types of ΔΔG values were plotted against the natural logarithm (ln) of % in vitro activity. Then, the values were analyzed and interpreted using Pearson’s correlation. The Pearson correlation coefficients (r) between three types of changes in free energy and the experimental activity of each mutant are −0.7019 (*p* = 0.0012), −0.7443 (*p* = 0.0004), and −0.5770 (*p* = 0.0122) for the ΔΔG binding, ΔΔG folding of the complex, and ΔΔG folding of the isolated ligand, as shown in Appendix A, respectively.

Additionally, we also obtained the equation for activity prediction according to the correlation between the experimental activity and three types of predicted free energy (ΔΔG) for the following:ΔΔG binding               :  Y = −0.6046x − 0.5622 ΔΔG folding of complex       :  Y = −0.4544x + 0.5580ΔΔG folding of isolated ligand  :  Y = −0.2829x + 0.5701where Y is the estimated the relative free energy (ΔΔG) value and X is the predicted experimental activity.

According to abovementioned results from the validation process, a significant association between the relative free energy (ΔΔG) and in vitro activity of IL-18 was observed. A higher ΔΔG reflected less activity. Therefore, this computational approach was considered as a potential method to obtain quantitatively reliable predicted free energy change (ΔΔG) upon mutation, so that it could be used for screening the favorable amino acid substitution of IL-18.

### 3.3. In Silico Screening of Favorable Mutation

To identify a potential candidate mutation, we considered the relative free energy change upon mutation in terms of ΔΔG_binding_, ΔΔG_folding of complex_, and ΔΔG_folding of isolated ligand_ values as the energy-based criterion for the selection of a favorable mutation. A designed mutation should improve the binding affinity to the receptor as well as preserve the overall conformation stability of IL-18 in both isolated and complex structures. The mutation(s) with favorable energetic contribution or ΔΔG < 0 are prone to be potential candidate mutations.

As an activity of IL-18 showed a good correlation with the relative free energy change (ΔΔG) upon mutation, we thus decided to rank the promising candidate mutation by using the following scoring scheme:

Mutation score = ∑ (ΔΔG_binding_ × w1) + (ΔΔG_folding of complex_ × w2) + (ΔΔG_folding of isolated ligand_ × w3).

According to the abovementioned equation, w1, w2, and w3 are coefficients, being −0.702, −0.744, and −0.577 for ΔΔG_binding_, ΔΔG_folding of complex_, and ΔΔG_folding of isolated ligand_, respectively. The higher score indicated promising design mutation.

#### 3.3.1. Single Point Mutation Study

The single point mutation(s) of the amino acid involved in the binding interface was considered as an effective strategy to improve the biological activity and stability of IL-18 [10,11,28]. In our work, the selected unfavorable residues of IL-18 involved in the direct binding to IL-18R were subjected to in silico saturation mutagenesis by mutating them to all other 19 amino acids.

Their impacts on IL-18 and IL-18R binding affinities, on IL-18 stability, and on the overall stability of the IL-18/IL-18R complex were then predicted using our validated computational protocols. Finally, eleven amino acid points were identified as unfavorable binding residues, namely, Glu6, Asn41, Arg44, Asp54, Gln56, Asp110, Asn111, Met113, Lys129, Arg131, and Asn155. The changes in free energy (ΔΔGs) were calculated; they are shown as the heat maps in Figure 2.

A total of 65 (31.1%) of the 209 mutations were predicted to increase affinity to the receptor with favorable binding free energies compared to the wild-type protein (Figure 2A). Considering variant stabilities, 40 (19.1%) mutations were predicted to stabilize the interface of the IL-18/IL-18R complex (Figure 2B) and 46 (22%) mutations tended to conserve the stability of the isolated IL-18 (Figure 2C) due to their favorable free energies compared to the wild-type. Based on the selection criteria, the E6M mutant was the best candidate for single point mutation (Table 1).

The E6M mutant was consistent with the previous in vitro studies, where the mutation of E6 to alanine and lysine (E6A and E6K) showed the increased activity of IL-18 compared to the wild-type [10]. These suggested that E6 is a promising target for designing mutation to increase IL-18 activity. However, IL-18 with T63A did not correlate with free energy change upon mutation. This allowed us to speculate that our protocol would work well for the case of amino acid(s) at the direct binding IL-18/IL-18R interface because all selected mutations, except T63A, are the amino acids directly contributing to binding.

#### 3.3.2. Double Mutation Study

In the case of Il-18, an increase in biological activity and therapeutic efficacy was observed toward the double mutation in the binding residues of IL-18 [9,10,14]. To predict the synergistic effects, all nine possible mutual combinations of the favorable single mutations were studied. Five single mutations, E6M, K129M, R131K, N111S, and R131G, were combined to produce nine double mutants.

Table 2 exhibits that all combinations of single favorable mutations were predicted to increase both the binding affinity (ΔΔG_binding_) and stability toward the receptor (ΔΔG_binding_) and protein itself (ΔΔG_folding of ligand_) compared to the wild-type. We found that four of the five best double mutants ranked by mutation score are composed of E6M. Interestingly, the best candidate was a single point mutation. Combinations of the E6K+T63A mutations could synergize the augmentation of biological function and clinical application of IL-18 in both in vitro and in vivo studies [9,14]. These results provided additional evidence to support the finding that the presence of the synergistic mutational effect of E6 significantly improved the IL-18 activity.

#### 3.3.3. Multiple Mutation Study

Nowadays, directed evolution techniques have emerged as a promising strategy for improving the functionality of proteins by increasing the mutation rate, resulting in the possibility of finding an optimal mutation [29]. Applying this strategy to our work, we decided to investigate the additive effects by introducing multiple mutations to IL-18. Five single mutations, E6M, K129M, R131K, N111S, and R131G, were combined into seven triple and two quadruple mutations. Predicted values of the changes in free energy (ΔΔGs) were calculated, and are summarized in Table 3.

The synergetic energetic effects of the predicted relative free energies (ΔΔGs) were observed from all variants with multiple mutations. Moreover, the candidate mutants with multiple mutations showed the best mutation score among the three mutation strategies, suggesting that the combination of multiple mutations of the IL-18 residue is considered as a potential strategy to improve their biological activity. The previous work reported that combination of S10T+D17N+T63A mutations exhibited a 2-fold greater activity compared to the wild-type [11]. Our computational strategy provided a benefit to identify favorable combinatorial mutations increasing the activity of IL-18.

#### 3.3.4. Selection of Favorable Mutations

As mentioned in the previous section, we proposed a computational strategy for identifying favorable mutations. Among all possible mutations, the best three candidate mutants, ranked by mutation score, showed the highest binding interaction and structural stabilization compared to other mutations, namely
-E6M+K129M+R131G+N111S;-E6M+N111S+R131G;-E6M+K129M+R131G.

These variants could be a promising candidate for improving protein activity. Moreover, our findings revealed synergistic mutational effects of the E6M mutant when combined with other mutations, suggesting that the modulation of IL-18 activity may have been influenced by the Glu6 residue. The 3D structure between IL-18 with the selected candidate residues and IL-18Rα is shown in Appendix A.

### 3.4. MD Simulation Analysis of Mutations

An atomistic understanding of our selected candidate mutations is essential to rationally guide the computational design of potent and effective IL-18 as a therapeutic candidate. In our work, we used molecular dynamics (MD) simulation to investigate the impacts of these mutations on the structural and functional implications of the protein.

#### 3.4.1. Conformational Stability Analysis

To study the dynamic stabilities of each simulated model, we performed MD simulations of four candidate mutants (E6M, E6M+N111S+R131G, E6M+K129M+R131G, and E6M+N111S+K129M+R131G) and the model structures with reported in vitro activities (wild-type, E6K, and M33Q) for 150 ns at 310 K under 1 atm pressure. The root mean square deviation (RMSD) values of all atoms in each model were plotted as depicted in Figure 3A.

The RMSD values of all studied systems continuously increased in the first 15 ns, and then fluctuated in the range of 1.5 Å during the simulation, suggesting that the systems reached an equilibrium. Thus, the MD trajectories from the last 90 ns were extracted for further analysis. We found that the average RMSD value of the IL-18 wild-type, E6M, E6M+N111S+R131G, E6M+K129M+R131G, E6M+N111S+K129M+R131G, E6K (positive control), and M33Q (negative control) were 1.78 ± 0.322, 1.25 ± 0.15, 1.50 ± 0.23, 1.34 ± 0.21, 1.97 ± 0.42, 1.54 ± 0.29, and 1.73 ± 0.39 Å, respectively.

#### 3.4.2. Structural Comparison

Moreover, we also analyzed the average distance of all backbone atoms in each amino acid from an origin point. The distance pattern was compared among the wild-type and mutant structures. A difference in the average distance of corresponding amino acids indicated the structural alternation at the atomistic level. The average distance of the geometric center of backbone atoms (N, C, Cα, and O atoms) to an origin point (0,0,0) in each simulated structure in angstrom (Å) units were calculated from the last 1000 MD trajectories.

The conformational comparison by distance pattern exhibited that there is no significant difference in the average distance of backbones atoms in each amino acid between the wild-type and mutant structures of IL-18 (Figure 3B). Taken together, the superimposition of the protein structures (Appendix A) also confirmed the structural conservation of our mutated IL-18 with respect to the native protein. The results demonstrated that the protein structure of the IL-18 wild-type and all mutants could be aligned through the backbone atoms with the generally acceptable range of the RMSD below 3 Å.

To better understand the effects of mutations on IL-18 dynamics, we calculated the per-residue RMSD and RMSF after the backbone alignment for all seven systems (WT, E6K, M33Q, and four designed variants (Appendix A)). The WT protein showed a stable fold with moderate flexibility in the β8–β9 hairpin loop (residues 106–112). The experimentally validated positive control E6K had globally higher per-residue RMSD values than the WT and M33Q, indicating broader conformational sampling, while maintaining moderate RMSF levels consistent with increased activity.

In comparison, the negative control M33Q had a similar global RMSD to the WT, but had a significant RMSF spike in the β8–β9 loop, indicating increased local mobility and decreased activity. Our in silico-designed variants (E6M, E6M+N111S+R131G, E6M+K129M+R131G, and E6M+N111S+K129M+R131G) maintained fold stability with a WT-like global RMSD, and showed only moderate increases in β8–β9 flexibility, avoiding the destabilizing dynamic profile of M33Q. The quadruple mutant exhibited the strongest, but still confined, loop fluctuations. This suggests that the controlled enhancement of β8–β9 loop mobility, combined with favorable energetic substitutions, may improve IL-18/IL-18Rβ engagement and biological activity.

In agreement with previous work, molecular dynamics and functional studies of IL-18 reported that the structural perseverance was primarily responsible for triggering the biological activity of the protein [9]. Therefore, these results suggested that our candidate mutations had no effect on the overall conformation of IL-18.

#### 3.4.3. Hydrogen Bond Analysis

Since the biological function of proteins depends on their stable configuration, hydrogen bonding interaction plays a fundamental role in stabilizing the three-dimensional structure of protein. To gain insight into the hydrogen bond details from the IL-18 wild-type and mutants, the number of intra-molecular hydrogen bonds in each protein structure were analyzed by using the final 90 ns MD trajectories according to the subsequent criteria: (1) a distance between proton donor (D) and acceptor (A) atoms of less than 3.5 Å, and (2) HD-H•••HA angle of at least 150°, as presented in Figure 3C.

We found that the average intra-molecular hydrogen bond of the IL-18 wild-type, E6M, E6M+N111S+R131G, E6M+K129M+R131G, E6M+N111S+K129M+R131G, E6K, and M33Q was 94, 88, 86, 86, 85, 85, and 84 H-bonds, respectively. These results showed a minimum difference in the hydrogen bond formation between the candidate mutant structures and the protein structures with experimentally reported activities (wild-type, E6K, and M33Q mutants).

These suggested that our candidate mutants were able to maintain their overall conformation through the stabilization from the intra-molecular hydrogen bond. However, the number of intra-molecular hydrogen bonds were observed to slightly decrease in all mutated IL-18 structures compared to the wild-type protein, suggesting that the alteration of amino acid might trigger a small conformational change in the individual region of IL-18.

#### 3.4.4. Structural Flexibility Analysis

Apart from the conformational stability, structural flexibility was observed to examine the effects associated with the mutations on the dynamic behavior of IL-18. The root mean square fluctuations (RMSFs) of all Cα atoms in each simulated structure were analyzed to identify the most flexible regions in the protein structure. The higher RMSF value indicates a more flexible region, whereas the lower RMSF value means a less flexible region. In this study, the RMSF values of each simulated structure were calculated using the final 90 ns MD trajectories, and were then plotted against the residue order of the protein, as shown in Figure 3D.

Comparison of the wild-type structure and the mutated structures of IL-18 exhibited that the RMSF profiles of the IL-18 mutants showed a similar pattern to that of the wild-type in most of the overall structure. However, we observed that the fluctuation in amino acid residues, from V106 to K112, of all mutated IL-18 structures were higher than the native protein structure, especially in the M33Q (negative control) and E6K (positive control) mutants. These results revealed the induction of the flexibility at the V106 to K112 residue could cause the alteration of the β8–β9 hairpin loop direction of the candidate mutants to provide closer contact with the IL-18Rβ, leading to the better binding affinity to its receptor (Figure 3D).

Moreover, these residues were also involved in binding site III of IL-18, required for the activation of the IL-18 function through the formation of a ternary complex with the receptors [7,8]. Therefore, our simulations suggested that the increased dynamics of the β8–β9 hairpin loop could have contributed to the biological activity of the IL-18 variants.

#### 3.4.5. Binding Interaction Analysis

Analysis of the binding interaction between the IL-18 mutants and IL-18R(s) was performed to elucidate the functional significance of the four mutation candidates in the context of receptor binding. The interaction of each simulated structure toward its receptors was analyzed and visualized using VMD software. The interaction profiles of the key residues at the interface between the IL-18R(s) and the candidate mutants, E6M, E6M+K129M+R131G, and E6M+N111S+R131G, are summarized in Appendix A, respectively.

The results exhibited the similarity between the binding interactions of four candidate mutants and the receptors. The E6M mutant revealed eight possible binding residues involved in receptor binding sites I, II, and III, including D17, D32, D40, K53, D110, K112, D132, and R147, while the E6M+K129M+R131G and E6M+N111S+R131G mutants exhibited nine binding residues, including D17, D32, D40, K53, H109, D110, K112, D132, and R147. Interestingly, the best candidate mutant, E6M+N111S+K129M+R131G, showed the most favorable binding interaction to its receptors via 10 binding residues, including D17, D32, D37, D40, K53, H109, D110, K112, D132, and R147.

Atomistic insights revealed that the key residues of the E6M+N111S+K129M+R131G mutant at the interface of the IL-18/IL-18R complex make closer contact to IL-18Rα (site I and II) and IL-18Rβ (site III), as shown in Figure 4 and Figure 5, respectively. Therefore, the results were consistent with the predicted free energy contribution, showing that our candidate mutants could improve the IL-18 receptor binding, and the main contributions to the binding interactions were from electrostatic interactions.

To strengthen our binding analysis, Figure 6 and Figure 7 demonstrate the electrostatic, long-range electrostatic, anion–π, and cation–π interactions at IL-18Rα (sites I–II) and IL-18Rβ (site III). Across representative snapshots (0, 50, 100, and 150 ns), these interactions remained within favorable ranges (≤4–5 Å for electrostatic and π interactions; >6 Å for long-range electrostatics), indicating that they are stable under MD conditions. In addition, time-dependent distance profiles confirmed that the contacts persisted throughout the 150 ns trajectories. Notably, these persistent interactions, including D37–R25, D40–H27, D146–Y214, K112–E210, and H109–E210, are not located directly on the mutated positions themselves. Instead, the designed substitutions (E6M, N111S, K129M, and R131G) appear to facilitate the electrostatic environment and backbone dynamics, indirectly stabilizing neighboring receptor–ligand interactions. This observation highlights the importance of non-hotspot mutations, which enhance protein–protein binding by modulating the interface landscape rather than by forming direct new contacts on the mutated residues.

This study used residue-level energy decomposition, in silico mutagenesis, and molecular dynamics (MD) simulations to find IL-18 variants with higher receptor binding potential. Our analysis identified the residue Glu6 as a major hotspot, with contributions from N111, K129, and R131. Per-residue decomposition identified these positions as energetically unfavorable due to electrostatic penalties or poor packing, providing justification for their replacement. Among single mutants, E6M was consistently the most favorable. Although E6M is a hydrophobic substitution, it is mechanistically consistent with the electrostatic improvements seen in our top multi-site designs. Glu6 causes an unfavorable electrostatic and desolvation penalty at the IL-18/IL-18R interface. Replacement with Met eliminates the penalty by removing the charged side chain and introducing hydrophobic packing, which stabilizes the local environment.

The per-residue RMSD and RMSF analyses (Appendix A) confirmed that E6M preserved the global fold but induced moderate increases in flexibility at the β8–β9 loop (residues 106–112), a region essential for IL-18Rβ engagement. This tuned flexibility differs from the destabilizing RMSF spike observed in the negative control M33Q and resembles the beneficial flexibility pattern of the positive control E6K, consistent with its experimentally validated enhancement of activity. Time-dependent interaction profiles (Figure 6 and Figure 7) further showed that E6M indirectly stabilized persistent electrostatic and π contacts (e.g., D37–R25, D40–H27, and D146–Y214), demonstrating that its effect extends beyond the mutation site by reshaping the interface environment.

The designed multi-site variants (E6M+N111S+R131G, E6M+K129M+R131G, and E6M+N111S+K129M+R131G) preserved fold stability (WT-like global RMSD) while consistently increasing flexibility in the β8–β9 loop. The quadruple mutant showed the strongest, but still confined, fluctuations, supporting the hypothesis that controlled loop mobility promotes IL-18Rβ engagement. Importantly, distance analyses, included in Figure 4 and Figure 5, provide direct evidence that electrostatic, long-range electrostatic, anion–π, and cation–π interactions at sites I–III remain stable across MD snapshots (0, 50, 100, and 150 ns).

Notably, most of these interactions did not occur directly on the mutated residues but were stabilized indirectly through the designed substitutions reshaping the electrostatic environment and backbone conformation. This observation is consistent with the concept of second-shell or non-hotspot mutations, which enhance protein–protein affinity by modifying the interface landscape rather than introducing direct new contacts. Together, these findings support a unified mechanistic interpretation: beneficial mutations act by removing unfavorable electrostatics, introducing stabilizing hydrophobic packing, and tuning β8–β9 loop dynamics to promote receptor engagement. The experimentally validated benchmarks further reinforce this model: E6K increased global conformational sampling without destabilization, M33Q introduced excessive loop flexibility and impaired activity, and our designed variants achieved an intermediate state of controlled, localized dynamics that favors binding.

However, we note several limitations: (i) our simulations were not performed on the full ternary IL-18/IL-18Rα/IL-18Rβ assembly that governs signaling; (ii) our predictions rely on computational analyses and require future experimental validation through mutagenesis and binding assays; and (iii) long-term stability and developability considerations, including aggregation propensity, immunogenicity, or neutralization by the IL-18 binding protein (IL-18BP), were not addressed. Future studies will extend our approach to the ternary complex and integrate experimental testing to confirm the predicted effects of these designed variants.

## 4. Conclusions

The idea of structure-guided protein design is to exploit the information from the detailed analysis of the structure, function, and protein–protein interactions at an atomistic level to design a new promising protein. In this study, the integration of computational structure-based energy calculation and molecular dynamic simulations were used to screen and identify candidate mutations, increasing the activity of IL-18 based on the hypothesis that favorable energetic contribution and structural conservation is needed for the receptor recognition of IL-18. Thus, the amino acid substitution of IL-18, which had a favorable energetic contribution and preserved the conformation of the protein compared to the native protein, may serve as a promising candidate for cytokine-based cancer drugs.

## Figures and Tables

**Figure 1 biomolecules-15-01360-f001:**
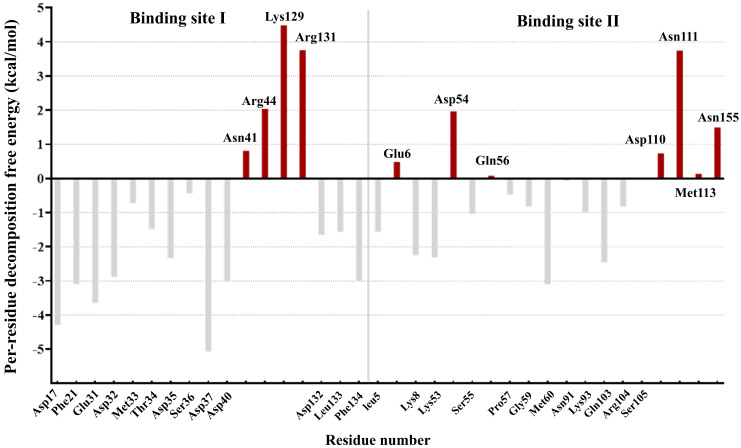
Per residue free energy contribution of IL−18 at the IL−18/IL−18R interface. The energy is in kcal/mol. The unfavorable interfacial residues of IL-18 that were selected for optimization are shown in red. The gray bars represent energetically favorable residues of IL-18 in binding sites I and II.

**Figure 2 biomolecules-15-01360-f002:**
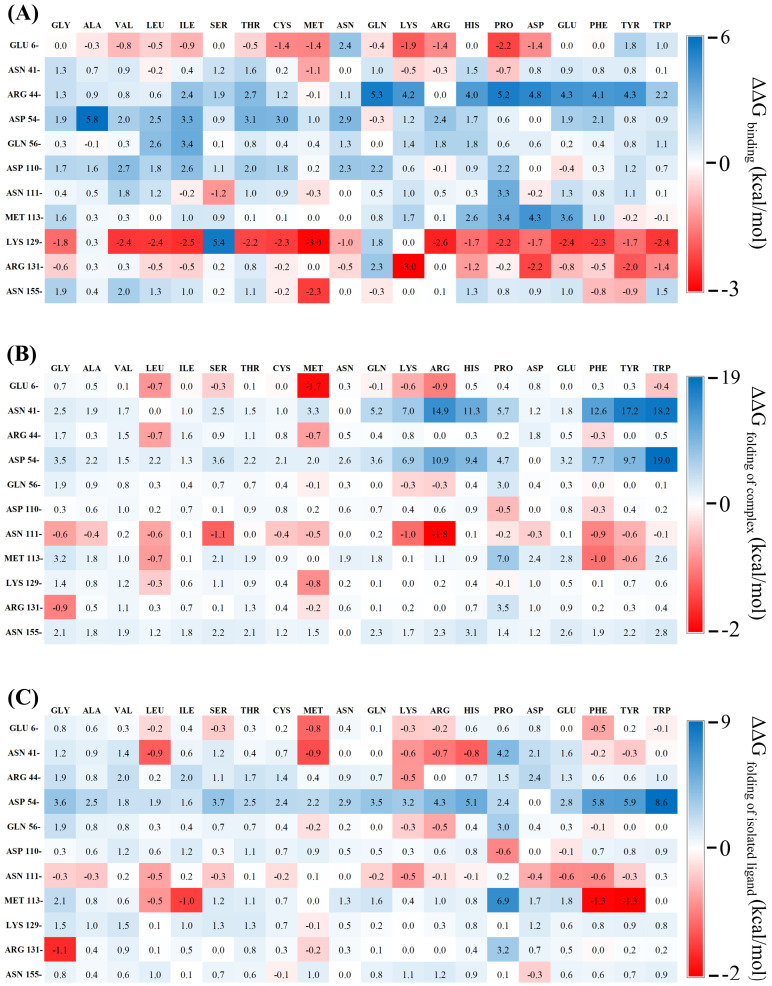
Heat maps showing predicted ΔΔG changes upon mutation of 11 unfavorable IL-18 residues. (**A**) ΔΔG_binding_: binding free energy change between wild-type and mutant IL-18 toward its receptor. (**B**) ΔΔG_folding of complex_: effect on IL-18/IL-18R complex stability. (**C**) ΔΔG_folding of isolated ligand_: effect on free IL-18 stability. Red and blue indicate increased/decreased values relative to wild-type. All values in kcal/mol.

**Figure 3 biomolecules-15-01360-f003:**
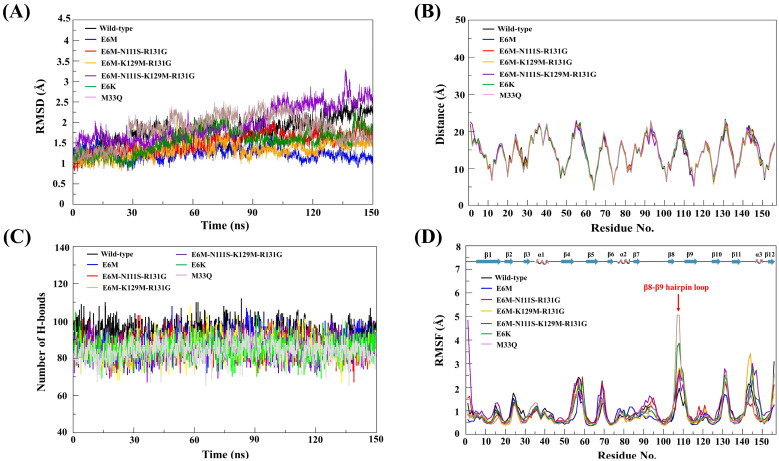
MD simulation profiles of IL-18 wild-type and mutant structures. (**A**) RMSD plots over 150 ns. (**B**) Average atomic distances. (**C**) Hydrogen bond profiles. (**D**) RMSF plots. Wild-type (black), E6M (blue), E6M-N111S-R131G (red), E6M-K129M-R131G (yellow), E6M-N111S-K129M-R131G (purple), E6K (green), and M33Q (gray).

**Figure 4 biomolecules-15-01360-f004:**
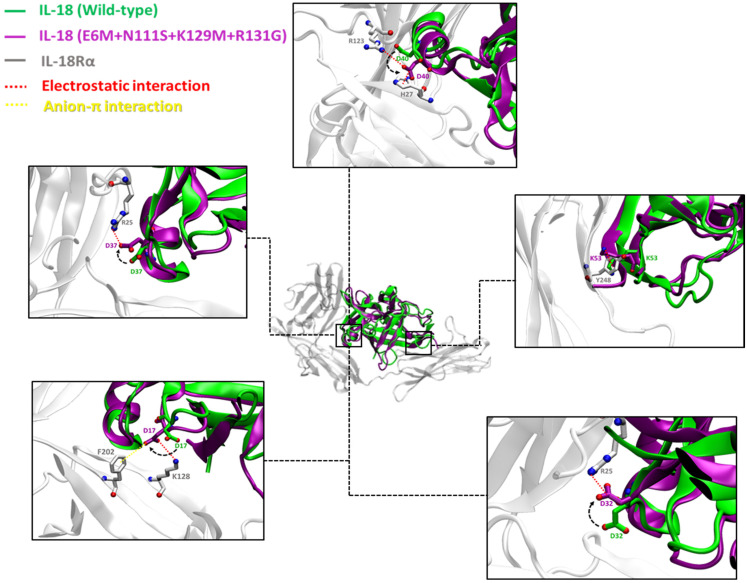
Binding interaction of IL-18 with E6M+N111S+K129M+R131G mutation (purple) and IL-18Rα (gray) at receptor binding sites I and II compared to wild-type (green). The electrostatic interactions are shown as red dashed lines, and the anion–π interaction is shown with a yellow dashed line.

**Figure 5 biomolecules-15-01360-f005:**
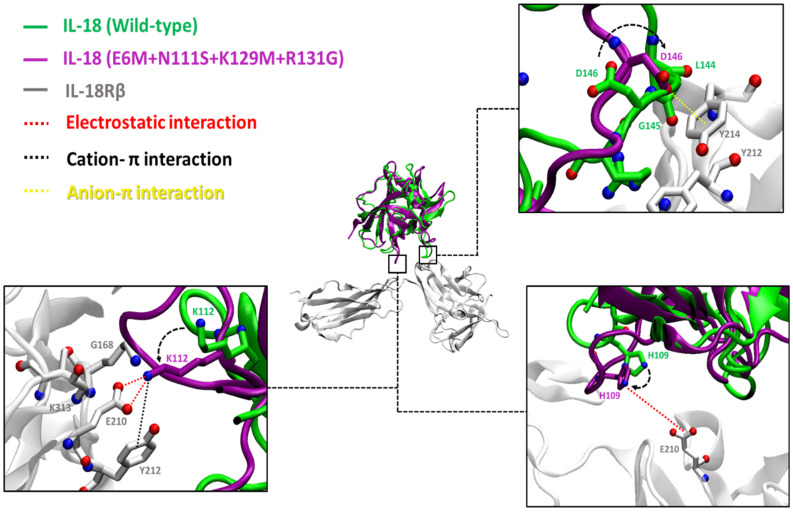
Binding interaction of IL-18 with E6M+N111S+K129M+R131G mutation (purple) and IL-18Rβ (gray) at receptor binding site III compared to wild-type (green). The electrostatic interactions are shown as red dashed lines, the cation–π interaction is shown with a black dashed line, and the anion–π interaction is shown with a yellow dashed line.

**Figure 6 biomolecules-15-01360-f006:**
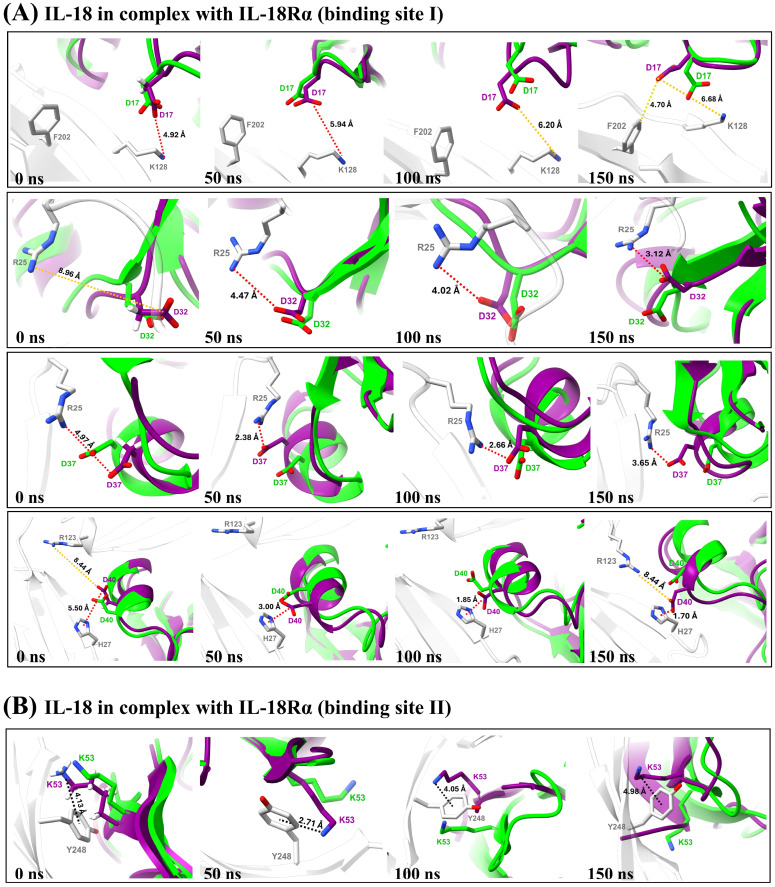
Time-dependent distances (Å) of key interactions at IL-18/IL-18Rα binding sites I and II for the E6M+N111S+K129M+R131G mutant over 150 ns MD.

**Figure 7 biomolecules-15-01360-f007:**
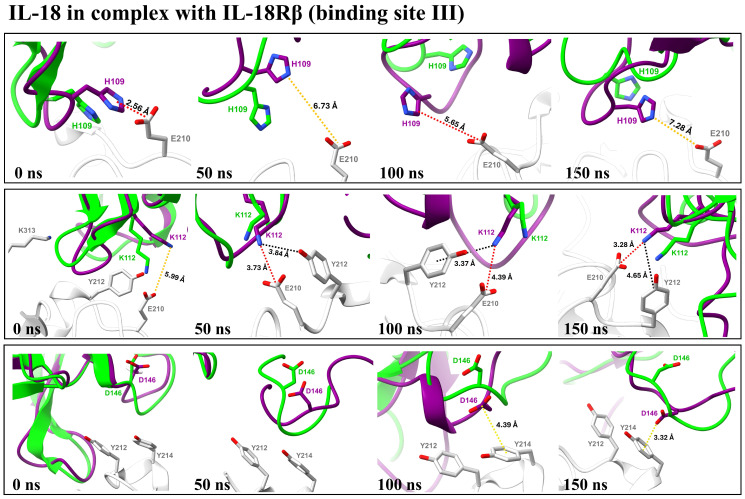
Time-dependent distances (Å) of key interactions at IL-18/IL-18Rβ binding site III for the E6M+N111S+K129M+R131G mutant over 150 ns MD.

**Table 1 biomolecules-15-01360-t001:** In silico predicted relative free energy (ΔΔG) of the best five variants in single point mutation study ranked by our mutation score. All ΔΔG values are in kcal/mol.

Mutation	ΔΔG_binding_	ΔΔG_folding of complex_	ΔΔG_folding of ligand_	Score
Control group				
Wild-type ^a^	0	0	0	0
E6K ^a^	−1.9	−0.6	0.3	1.95
T63A ^a^	−0.8	0.8	0.8	−0.44
M60Q ^a^	4.6	2.2	0.8	−5.35
M33Q ^a^	4.6	2.9	2.7	−7.02
Candidate group				
E6M	−1.4	−1.7	−0.8	2.76
K129M	−3.0	−0.8	−0.1	2.74
R131K	−3.0	0.2	0.0	1.93
N111S	−1.2	−1.1	−0.3	1.87
R131G	−0.6	−0.9	−1.1	1.70

^a^ Biological activity of wild-type and E6K (+ve), M33Q (−ve), and M60Q (−ve) mutants were evaluated experimentally by IFN-γ induction assay in NK-92MI cells when compared to IL-18 wild-type [9].

**Table 2 biomolecules-15-01360-t002:** In silico predicted relative free energy (ΔΔG) of the variants in double mutation study ranked by our mutation score. All ΔΔG values are in kcal/mol.

Mutation	ΔΔG_binding_	ΔΔG_folding of complex_	ΔΔG_folding of ligand_	Score
Control group				
Wild-type ^a^	0	0	0	0
E6K ^a^	−1.9	−0.6	0.3	1.95
T63A ^a^	−0.8	0.8	0.8	−0.44
M60Q ^a^	4.6	2.2	0.8	−5.35
M33Q ^a^	4.6	2.9	2.7	−7.02
Candidate group				
K129M+R131G	−2.0	−2.5	−2.1	4.48
E6M+R131G	−1.9	−2.6	−1.9	4.43
E6M+K129M	−2.3	−2.6	−0.9	4.02
E6M+N111S	−2.0	−2.6	−1.1	3.98
E6M+R131K	−3.4	−1.5	−0.7	3.94
N111S+R131G	−2.2	−1.9	−1.4	3.82
R131K+N111S	−4.0	−0.9	−0.2	3.63
K129M+N111S	−2.6	−2.0	−0.4	3.52
K129M+R131K	−2.4	−0.6	0.0	2.09

^a^ Biological activity of wild-type and E6K (+ve), E6K+T63A (+ve), M33Q (−ve), and M60Q (−ve) mutants were evaluated experimentally by IFN-γ induction assay in NK-92MI cells when compared to IL-18 wild-type [9].

**Table 3 biomolecules-15-01360-t003:** In silico predicted relative free energy (ΔΔG) of the variants in multiple mutation study ranked by our mutation score. All ΔΔG values are in kcal/mol.

Mutation	ΔΔG_binding_	ΔΔG_folding of complex_	ΔΔG_folding of ligand_	Score
Control group				
Wild-type ^a^	0	0	0	0
E6K ^a^	−1.9	−0.6	0.3	1.95
T63A ^a^	−0.8	0.8	0.8	−0.44
M60Q ^a^	4.6	2.2	0.8	−5.35
M33Q ^a^	4.6	2.9	2.7	−7.02
Candidate group				
E6M+K129M+R131G+N111S	−5.4	−5.6	−3.2	9.78
E6M+N111S+R131G	−4.8	−4.0	−2.2	7.66
E6M+K129M+R131G	−3.7	−4.3	−2.9	7.45
K129M+N111S+R131G	−4.7	−3.8	−2.3	7.45
E6M+K129M+R131K+N111S	−4.8	−3.1	−1.1	6.35
E6M+R131K+N111S	−4.2	−2.3	−1.0	5.27
E6M+K129M+N111S	−2.3	−3.5	−1.2	4.88
E6M+K129M+R131K	−3.5	−2.3	−0.8	4.65
K129M+R131K+N111S	−3.9	−1.7	−0.3	4.13

^a^ Biological activity of wild-type and E6K (+ve), E6K+T63A (+ve), M33Q (−ve) and M60Q (−ve) mutants were evaluated experimentally by IFN-γ induction assay in NK-92MI cells when compared to IL-18 wild-type [9].

## Data Availability

The original contributions presented in this study are included in the article/Appendix A. Further inquiries can be directed to the corresponding author.

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
