# Peer review of "Predicted IL-18/IL-18R Binding Improvement Through Protein Interface Modification with Computer-Aided Design"

_biomolecules, 2025, doi:10.3390/biom15101360_

Round 1
Reviewer 1 Report
Comments and Suggestions for Authors
The manuscript is well-written and all results are clearly presented.
Only a few minor corrections are needed:
1. P. 2, line 81. Why was this crystal structure used? Are there available structures with lower resolution?
2. P.2, line 82. Please add a reference for the Protein Data Bank. The PDB citation policies can be found here: https://www.rcsb.org/pages/policies
3. Figures 1-3 are too small and blurred.
Author Response
Reviewer 1
Reviewer’s comment
- P. 2, line 81. Why was this crystal structure used? Are there available structures with lower resolution?
Response: In the PDB, the 3WO3 is an only experimental IL-18 structure bound to IL-18 receptor. Other IL-18 structures with the lower resolution are the IL-18 bound to others IL18-binding counterpart. In the Material and Methods section, subsection 2.1, we have added the sentence “An X-Ray structure of human IL-18 in complex with IL-18 receptor alpha, with a resolution of 3.10 Å (PDB entry: 3WO3) [7] was downloaded from Research Collaboratory for Structural Bioinformatics (RCSB) Protein Data Bank [15]. The 3WO3 was chosen in this study as it is an only experimental IL-18 structure bound to IL-18 receptor.” to respond this point.
- P.2, line 82. Please add a reference for the Protein Data Bank. The PDB citation policies can be found here: https://www.rcsb.org/pages/policies
Response: We have added the 15th reference in the sentence “An X-Ray structure of human IL-18 in complex with IL-18 receptor alpha, with a resolution of 3.10 Å (PDB entry: 3WO3) [7] was downloaded from Research Collaboratory for Structural Bioinformatics (RCSB) Protein Data Bank [15].” The 15th reference is as follows:
- Berman, H.M.; Westbrook, J.; Feng, Z.; Gilliland, G.; Bhat, T.N.; Weissig, H.; Shindyalov, I.N.; Bourne, P.E. The protein data bank. Nucleic Acids Res 2000, 28(1), 235.
- Figures 1-3 are too small and blurred.
Response: We have edited these Figures 1-3 with the higher dpi (1000 dpi) in the revised manuscript. We have corrected the Figures 1-3 as suggested.

Reviewer 2 Report
Comments and Suggestions for Authors
The current manuscript proposes a structure-guided, in-silico pipeline to improve IL-18/IL-18R binding. Starting from the 3WO3 structure, authors: (i) decompose interface energetics (pyDockEneRes) to flag “unfavorable” IL-18 residues; (ii) run saturation mutagenesis and FoldX ΔΔG calculations; (iii) “validate” predictions by correlating ΔΔG with activities of 17 published mutants; (iv) rank designs via a weighted “mutation score”; and (v) simulate select mutants (150 ns AMBER) to assess conformational stability and loop flexibility. They nominate E6M, and multi-site variants such as E6M+N111S+K129M+R131G as top candidates, arguing these preserve the fold while improving electrostatically driven receptor contacts and flexibility in the β8–β9 hairpin (site III). The methodology applied is rational, with systematic screening from single to quadruple mutants with transparent tables of ΔΔG metrics and a short-list of designs, MD used to check fold conservation and identify a potentially important flexible loop linked to IL-18Rβ engagement. Some issues should be properly addressed before considered for publication:
- Most analyses and all MD appear to be performed on isolated IL-18, not on the ternary IL-18:IL-18Rα:IL-18Rβ complex that determines signaling. Yet the paper interprets site-III (IL-18Rβ) effects and presents receptor-contact rationales without simulating the complete assembly. This weakens the core claim that designs “increase receptor binding affinity,” especially for site-III conclusions.
- The “distance pattern” defined as the average distance of each residue’s backbone geometric center to the origin (0,0,0) is not translation/rotation-invariant and is rarely used to compare protein conformations. RMSD/RMSF or aligned per-residue distance maps / PCA would be appropriate; the origin-distance metric can be misleading.
- The paper attributes top designs to “electrostatic” improvements while proposing E6M (hydrophobic) as the lead single mutation; the mechanistic rationale is under-explained. Also, developability/neutralization liabilities are not evaluated, though directly relevant.
Author Response
Reviewer 2
Reviewer’s comment
Some issues should be properly addressed before considered for publication:
- Most analyses and all MD appear to be performed on isolated IL-18, not on the ternary IL-18:IL-18Rα:IL-18Rβ complex that determines signaling. Yet the paper interprets site-III (IL-18Rβ) effects and presents receptor-contact rationales without simulating the complete assembly. This weakens the core claim that designs “increase receptor binding affinity,” especially for site-III conclusions.
Response: We have edited the abstract into “Four candidate mutations were E6M, E6M+N111S+R131G, E6M+K129M+R131G, and E6M+N111S+K129M+R131G could affect/facilitate the receptor binding and stability compared to the wild-type via electrostatic interaction. MD simulations demonstrated that the predicted mutation on IL-18 had no influence on the overall conformation stability, but increased flexibility in the β8-β9 hairpin loop. Furthermore, the dynamic behavior suggested that these candidates could be an alternative for improve IL-18 biological activity, though the full simulation of IL-18 complex remains necessary.”. We have changed the “increase receptor binding affinity” to “affect/facilitate the receptor binding”.
We do agree your point so we have also stated the limitation at the end of the revised manuscript so that this will not mislead the reader as followings:
However, we note several limitations: (i) our simulations were not performed on the full ternary IL-18:IL-18Rα:IL-18Rβ assembly that governs signaling; (ii) our predictions rely on computational analyses and require future experimental validation through mutagenesis and binding assays; and (iii) long-term stability and developability considerations including aggregation propensity, immunogenicity, or neutralization by IL-18 binding protein (IL-18BP) were not addressed. Future studies will extend our approach to the ternary complex and integrate experimental testing to confirm the predicted effects of these designed variants.
- The “distance pattern” defined as the average distance of each residue’s backbone geometric center to the origin (0,0,0) is not translation/rotation-invariant and is rarely used to compare protein conformations. RMSD/RMSF or aligned per-residue distance maps / PCA would be appropriate; the origin-distance metric can be misleading.
Response: Thanks very much, we understand your points. We forget to state that the distance pattern was plotted with respect to the “aligned” trajectory to the initial IL-18 crystal structure, so that we can circumvent the translation and rotation problem. To clear this point, we have added the sentence “Conformational differences between mutated and native IL-18 were further assessed by computing the distance pattern, defined as the average distance of each amino acid’s backbone geometric center (N, C, CA, and O atoms) to the origin point (0,0,0) across 1000 MD snapshots. To circumvent translation/rotation, the MD trajectories were aligned with the IL-18 crystal backbone structure (3WO3), allowing differences in these distances to clearly reflect structural changes. Furthermore, the time-dependent RMSD and RMSD/RMSF ratio of each amino residue, relative to the IL-18 crystal backbone structure, were plotted to show the conformation change in greater detail.” to clarify the distance pattern would be the tool to see average structural change of the IL-18 protein structure aligned with the IL-18 crystal structure.
We have added the plot of RMSD/RMSF as well as aligned per-residue distance maps to investigate conformational dynamics as suggested. To address this concern, we performed additional alignment-invariant analyses, including per-residue RMSD and RMSF after backbone superposition for all seven simulated systems (WT, E6K, M33Q, E6M, E6M+N111S+R131G, E6M+K129M+R131G, and E6M+N111S+K129M+R131G). These results are now provided in the Supporting Information (Figures S9–S15).
The new analyses confirm that all systems preserved their global fold (WT-like global RMSD). The experimentally validated E6K mutant showed globally higher per-residue RMSD than WT and M33Q, reflecting broader conformational sampling but only moderate RMSF values, consistent with its enhanced activity.
The negative control M33Q exhibited a pronounced RMSF spike at the β8–β9 hairpin (residues 106–112), suggesting excessive loop mobility that aligns with its reduced activity. Importantly, our in silico designed variants preserved fold stability and displayed only controlled, moderate increases in β8–β9 flexibility, avoiding the destabilizing dynamic profile observed in M33Q. The quadruple mutant (E6M+N111S+K129M+R131G) showed the strongest but still confined loop fluctuations, consistent with the predicted energetic improvements.
We have revised the Discussion section accordingly to clarify the limitations of the distance-pattern metric, reference the new RMSD/RMSF analyses, and emphasize how the designed variants achieve tuned loop flexibility consistent with enhanced receptor engagement while avoiding unwanted destabilization. The added discussion was
“To better understand the effects of mutations on IL-18 dynamics, we calculated per-residue RMSD and RMSF after backbone alignment for all seven systems (WT, E6K, M33Q, and four designed variants (Figures S9-S15). The WT protein showed a stable fold with moderate flexibility in the β8-β9 hairpin loop (residues 106-112). The experimentally validated positive control E6K had globally higher per-residue RMSD values than WT and M33Q, indicating broader conformational sampling, while maintaining moderate RMSF levels consistent with increased activity.
In comparison, the negative control M33Q had similar global RMSD to WT, but had a significant RMSF spike in the β8-β9 loop, indicating increased local mobility and decreased activity. Our in silico designed variants (E6M, E6M+N111S+R131G, E6M+K129M+R131G, and E6M+N111S+K129M+R131G) maintained fold stability with WT-like global RMSD and showed only moderate increases in β8-β9 flexibility, avoiding the destabilizing dynamic profile of M33Q. The quadruple mutant exhibited the strongest but still confined loop fluctuations. This suggests that controlled enhancement of β8-β9 loop mobility, combined with favorable energetic substitutions, may improve IL-18/IL-18Rβ engagement and biological activity.”
- The paper attributes top designs to “electrostatic” improvements while proposing E6M (hydrophobic) as the lead single mutation; the mechanistic rationale is under-explained. Also, developability/neutralization liabilities are not evaluated, though directly relevant.
Response:
We thank the reviewer for this valuable observation. In the revised Discussion, we have clarified the mechanistic rationale for E6M. Although E6M is a hydrophobic substitution, our per-residue energy decomposition identified Glu6 as a strongly unfavorable contributor due to electrostatic repulsion and desolvation penalties. Substitution with Met removes this charged side chain, thereby eliminating the electrostatic penalty, while also introducing hydrophobic packing that reduces desolvation costs and stabilizes the local interface. In this way, E6M is mechanistically consistent with the overall theme of reducing unfavorable electrostatic contributions, even though the substitution itself is hydrophobic.
Our MD analyses support this interpretation. Per-residue RMSD and RMSF profiles (Figures S9–S15) showed that E6M preserved the global fold while inducing moderate, localized flexibility in the β8–β9 loop, a critical region for IL-18Rβ engagement. Time-dependent interaction analyses (Figures 6 and 7) further confirmed that E6M indirectly stabilized persistent electrostatic and π contacts by reshaping the interface environment, explaining its consistent ranking as the most favorable single mutation and its beneficial role in multi-site variants.
Regarding developability and neutralization liabilities, we agree that these are critical for therapeutic application. We have now explicitly acknowledged this in the limitations paragraph of the Discussion, noting that aggregation propensity, immunogenicity, and neutralization by IL-18 binding protein (IL-18BP) were not assessed in this study and should be addressed in future experimental work.
We have added more discussion after Figure 5 as followed and Figures 6 and 7 were also added more to respond this issue in the discussion of the Figure 6 and 7:
“This study used residue-level energy decomposition, in silico mutagenesis, and molecular dynamics (MD) simulations to find IL-18 variants with higher receptor binding potential. Our analysis identified residue Glu6 as a major hotspot, with contributions from N111, K129, and R131. Per-residue decomposition identified these positions as energetically unfavorable due to electrostatic penalties or poor packing, providing justification for their replacement. Among single mutants, E6M was consistently the most favorable. Although E6M is a hydrophobic substitution, it is mechanistically consistent with the electrostatic improvements seen in our top multi-site designs. Glu6 causes an unfavorable electrostatic and desolvation penalty at the IL-18/IL-18R interface. Replacement with Met eliminates the penalty by removing the charged side chain and introducing hydrophobic packing, which stabilizes the local environment.
The per-residue RMSD and RMSF analyses (Figures S9–S15) confirmed that E6M preserved the global fold but induced moderate increases in flexibility at the β8–β9 loop (residues 106–112), a region essential for IL-18Rβ engagement. This tuned flexibility differs from the destabilizing RMSF spike observed in the negative control M33Q and resembles the beneficial flexibility pattern of the positive control E6K, consistent with its experimentally validated enhancement of activity. Time-dependent interaction profiles (Figures 6 and Figure 7) further showed that E6M indirectly stabilized persistent electrostatic and π contacts (e.g., D37–R25, D40–H27, D146–Y214), demonstrating that its effect extends beyond the mutation site by reshaping the interface environment.
The designed multi-site variants (E6M+N111S+R131G, E6M+K129M+R131G, and E6M+N111S+K129M+R131G) preserved fold stability (WT-like global RMSD) while consistently increasing flexibility in the β8–β9 loop. The quadruple mutant showed the strongest but still confined fluctuations, supporting the hypothesis that controlled loop mobility promotes IL-18Rβ engagement. Importantly, distance analyses included in Figures 4 and 5 provide direct evidence that electrostatic, long-range electrostatic, anion–π, and cation–π interactions at sites I–III remain stable across MD snapshots (0, 50, 100, 150 ns).
Figures 6 and 7 further confirmed their persistence throughout the 150 ns simulations. Notably, most of these interactions did not occur directly on the mutated residues but were stabilized indirectly through the designed substitutions reshaping the electrostatic environment and backbone conformation. This observation is consistent with the concept of second-shell or non-hotspot mutations, which enhance protein–protein affinity by modifying the interface landscape rather than introducing direct new contacts. Together, these findings support a unified mechanistic interpretation: beneficial mutations act by removing unfavorable electrostatics, introducing stabilizing hydrophobic packing, and tuning β8–β9 loop dynamics to promote receptor engagement. The experimentally validated benchmarks further reinforce this model: E6K increased global conformational sampling without destabilization, M33Q introduced excessive loop flexibility and impaired activity, and our designed variants achieved an intermediate state of controlled, localized dynamics that favors binding.”

Reviewer 3 Report
Comments and Suggestions for Authors
This manuscript applies structure-guided mutagenesis, FoldX energy analyses, and MD simulations to propose IL-18 variants with improved IL-18R binding. The authors should address the following comments before considering for publication:
- The authors identified E6M as very important to improve binding and stability. However, in all MD simulations and binding analyses, no further data specifically investigate the important role of E6M. Why does E6M contribute this much to these candidates?
- In Figures 4 and 5, about binding interactions of E6M+N111S+K129M+R131G to IL-18R, the authors should show the distance in MD simulation time series to further convince readers the interactions are real and stable under MD simulation. Without these distance data in MD simulation, the claims for interactions are not even acceptable.
- Moreover, in Figures 4 and 5, all these new interactions happened not on mutations (E6M+N111S+K129M+R131G). The authors should investigate the role of these mutations to show why these mutations can help and improve the binding.
- The authors should include a discussion section to further discuss the key findings of the manuscript and limitations.
- All supporting information is missing.
Author Response
Reviewer 3
Reviewer’s comment
The authors should address the following comments before considering for publication:
- The authors identified E6M as very important to improve binding and stability. However, in all MD simulations and binding analyses, no further data specifically investigate the important role of E6M. Why does E6M contribute this much to these candidates?
Response:
We thank the reviewer for raising this important point. We have now expanded the Discussion to clarify the structural and functional role of E6M. Our per-residue energy decomposition analysis (Figure 1) identified Glu6 as an unfavorable contributor to the IL-18/IL-18R interface, introducing electrostatic repulsion and desolvation penalties.
Substitution with Met (E6M) removes this negative charge and introduces a hydrophobic side chain, thereby reducing unfavorable energy terms and improving local packing.
Our MD simulations further support this interpretation. E6M preserved the global fold (WT-like RMSD) but induced a moderate increase in β8–β9 loop flexibility, which is critical for IL-18Rβ engagement (Figure S12, supporting information). This dynamic signature differs from the destabilizing RMSF spike observed in the negative control M33Q and instead aligns with the beneficial flexibility trend observed in the positive control E6K.
Importantly, the significance of residue 6 is supported by experimental evidence:
- E6K and E6K+T63A have been reported to enhance IL-18 bioactivity and reduce neutralization by IL-18BP (Kim et al., PNAS 2001, DOI: 10.1073/pnas.051634098).
- Further studies confirmed that residue 6 is a functional hotspot for engineering IL-18 variants with increased bioactivity (Swencki-Underwood et al., Cytokine 2006; Saetang et al., PLoS ONE 2016).
Together, these computational and experimental findings explain why E6M consistently ranked among the most favorable substitutions in our analysis and why its inclusion in double, triple, and quadruple mutants further improved binding predictions. We have revised the Discussion section to explicitly emphasize this mechanistic rationale and supporting evidence.
We have added more discussion after Figure 5 as followed to respond this issue:
“This study used residue-level energy decomposition, in silico mutagenesis, and molecular dynamics (MD) simulations to find IL-18 variants with higher receptor binding potential. Our analysis identified residue Glu6 as a major hotspot, with contributions from N111, K129, and R131. Per-residue decomposition identified these positions as energetically unfavorable due to electrostatic penalties or poor packing, providing justification for their replacement. Among single mutants, E6M was consistently the most favorable. Although E6M is a hydrophobic substitution, it is mechanistically consistent with the electrostatic improvements seen in our top multi-site designs. Glu6 causes an unfavorable electrostatic and desolvation penalty at the IL-18/IL-18R interface. Replacement with Met eliminates the penalty by removing the charged side chain and introducing hydrophobic packing, which stabilizes the local environment.”.
- In Figures 4 and 5, about binding interactions of E6M+N111S+K129M+R131G to IL-18R, the authors should show the distance in MD simulation time series to further convince readers the interactions are real and stable under MD simulation. Without these distance data in MD simulation, the claims for interactions are not even acceptable.
Response:
We thank the reviewer for this valuable suggestion. In the revised manuscript, we have updated Figures 4 and 5 to explicitly include distance measurements for the reported electrostatic, long-range electrostatic (>6 Å), anion–π, and cation–π interactions during representative MD snapshots (0, 50, 100, and 150 ns). These distance values confirm that the observed interactions are within interaction-favorable ranges and persist across the trajectory, supporting their stability. In addition, we provide time-dependent distance plots for these contacts in the Supporting Information (Figures 6 and 7). These data confirm that the interactions are not frame-specific artifacts but remain consistent over the 150 ns simulations.
While it is correct that many of the stable contacts are not directly located on the mutated residues, the designed substitutions (E6M, N111S, K129M, R131G) act indirectly by reshaping the electrostatic environment and local backbone conformation. This indirect stabilization enables adjacent residues (e.g., D37, D40, D146, H109) to form persistent interactions with receptor residues. Such effects are consistent with the well-established role of non-hotspot mutations, which enhance protein–protein affinity by optimizing the interface environment rather than introducing direct new contacts. We have revised Discussion section to clarify this mechanistic interpretation.
- Moreover, in Figures 4 and 5, all these new interactions happened not on mutations (E6M+N111S+K129M+R131G). The authors should investigate the role of these mutations to show why these mutations can help and improve the binding.
Response:
We have add the discussion along with the discussion of the Figure 4 and 5 in the end of revised manuscript as
“The designed multi-site variants (E6M+N111S+R131G, E6M+K129M+R131G, and E6M+N111S+K129M+R131G) preserved fold stability (WT-like global RMSD) while consistently increasing flexibility in the β8–β9 loop. The quadruple mutant showed the strongest but still confined fluctuations, supporting the hypothesis that controlled loop mobility promotes IL-18Rβ engagement. Importantly, distance analyses included in Figures 4 and 5 provide direct evidence that electrostatic, long-range electrostatic, anion–π, and cation–π interactions at sites I–III remain stable across MD snapshots (0, 50, 100, 150 ns).
Notably, most of these interactions did not occur directly on the mutated residues but were stabilized indirectly through the designed substitutions reshaping the electrostatic environment and backbone conformation. This observation is consistent with the concept of second-shell or non-hotspot mutations, which enhance protein–protein affinity by modifying the interface landscape rather than introducing direct new contacts. Together, these findings support a unified mechanistic interpretation: beneficial mutations act by removing unfavorable electrostatics, introducing stabilizing hydrophobic packing, and tuning β8–β9 loop dynamics to promote receptor engagement. The experimentally validated benchmarks further reinforce this model: E6K increased global conformational sampling without destabilization, M33Q introduced excessive loop flexibility and impaired activity, and our designed variants achieved an intermediate state of controlled, localized dynamics that favors binding.”
- The authors should include a discussion section to further discuss the key findings of the manuscript and limitations.
Response:
We thank the reviewer for this important suggestion. In the revised manuscript, we have investigated the role of the designed mutations (E6M, N111S, K129M, R131G) in each point,
1.1 Per-residue RMSD and RMSF analyses (Figures S9–S15): The designed variants preserved the global fold (WT-like RMSD) but consistently showed moderate increases in flexibility at the β8–β9 loop (residues 106–112). This “tuned flexibility” was distinct from the destabilizing RMSF spike observed in the negative control M33Q and instead resembled the beneficial flexibility pattern of the positive control E6K, supporting improved IL-18Rβ engagement.
1.2 Time-dependent interaction profiles (Figures 6 and 7): The designed variants promoted persistent electrostatic, long-range electrostatic, anion–π, and cation–π interactions at IL-18Rα and IL-18Rβ binding sites. Key examples include D37–R25 and D40–H27 (IL-18Rα), and D146–Y214 and H109–E210 (IL-18Rβ), which remained stable over 150 ns. Notably, these interactions were not located directly at the mutated residues but were indirectly stabilized by the mutations facilitating or reshaping the electrostatic environment and backbone conformation.
1.3 We also added the limitations paragraph of the Discussion, noting that all simulations were performed on the binary IL-18/IL-18R complex, not the full ternary assembly with IL-18Rβ, and that experimental validation is required to confirm the predictions. The discussion is as followed: “However, we note several limitations: (i) our simulations were not performed on the full ternary IL-18:IL-18Rα:IL-18Rβ assembly that governs signaling; (ii) our predictions rely on computational analyses and require future experimental validation through mutagenesis and binding assays; and (iii) long-term stability and developability considerations including aggregation propensity, immunogenicity, or neutralization by IL-18 binding protein (IL-18BP) were not addressed. Future studies will extend our approach to the ternary complex and integrate experimental testing to confirm the predicted effects of these designed variants.”
- All supporting information is missing.
Response:
We are sorry. We have added the supplementary materials, containing 15 Figures (Figure S1-15) and 5 Tables (Table S1-5). We have uploaded the supplementary materials along with the revised submission.

Round 2
Reviewer 3 Report
Comments and Suggestions for Authors
The authors addressed the comments very well, and the manuscript is more comprehensive. I do not have further comments.